# How Do Mobile Wallets Improve Sustainability in Payment Services? A Comprehensive Literature Review

**Egemen Hopalı** [1,*], **Özalp Vayvay** [2], **Zeynep Tuğçe Kalender** [3,4], **Deniz Turhan** [5] and **Ceyda Aysuna** [6]

1 Department of Engineering Management, Institute of Pure and Applied Sciences, Marmara University, 34722 Istanbul, Turkey

2 Department of Industrial Engineering, Faculty of Engineering and Natural Sciences, Istanbul Health and Technology University, 06800 Istanbul, Turkey

3 Department of Enterprise Management and Economics, Faculty of Mechanical Engineering, Czech Technical University, 16000 Prague, Czech Republic

4 Department of Industrial Engineering, Faculty of Engineering, Marmara University, 34722 Istanbul, Turkey

5 Department of Industrial Engineering, Faculty of Engineering and Architecture, Fenerbahce University, 34758 Istanbul, Turkey

6 Department of Marketing, Faculty of Business Administration, Marmara University, 34722 Istanbul, Turkey

* Correspondence: e.hopali@gmail.com

**Abstract:** Easy access to the Internet, smartphones, and mobile-based banking change customer shopping intentions. As a crucial component of financial technology (Fintech), mobile wallets enable customers to shop via smartphones. Mobile wallets present a cashless transactional method, cost-efficient services, and traceable options that improve sustainability in payment services. Over the last decade, mobile wallet services have evolved and attracted considerable attention from customers and companies. Due to the need for a comprehensive mobile wallet literature survey, this article aims at filling this research gap by covering articles published between 2012 and 2022 over the Scopus, Web of Science, and Science Direct databases. A clear filtering policy was conducted to observe the related article topics. Thus, 128 articles that met the inclusion and exclusion criteria were analyzed. Moreover, the articles were initially classified into three main groups, which was performed via scanning and categorizing all studies in the last ten years from different databases. In addition, the literature was systematically reviewed, providing a better understanding of mobile wallets and contributing to the literature by researching how this service can be improved for payment services with a focus on sustainability. The conducted literature review revealed that mobile wallets could be promoted in terms of environmental traceability, customer lifetime value, and security.

**Keywords:** mobile wallet; e-wallet; sustainability; literature review

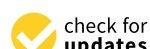



## 1. Introduction

The speed and diversity of technology-enabled innovations have increased over the last decade. In terms of sustainability, the benefits of green technology are significant for achieving ecological goals. According to the European Commission [1], reducing greenhouse gas emissions by at least 55% by 2030 compared to 1990 is a target needed to be reached for sustainability.

Traditional checkout payment methods, such as cash and plastic credit cards, emit an average of 3.78 g of $CO_2$ per transaction [2]. Considering this situation, a technology that can provide a cardless, traceable, and green payment service is vital for sustainability. One of the emerging technologies, Fintech, can be defined as technology-enabled innovation in financial services. Fintech is accepted as an alternative to traditional finance and improves sustainability in payment services with its components that offer paperless and traceable payment features. It employs innovative technologies such as data science, artificial intelligence, sensors, mobile technology infrastructure, advanced business operations,

Internet-based platforms, and big data analysis applications, which propel Fintech services [3,4]. A few trends empower its progression. Firstly, customers and merchants search for alternatives because of a lack of trust and confidence in incumbent service providers. Secondly, emerging financial services such as mobile wallets, peer-to-peer money transfers, and e-insurance offer innovative, efficient, and convenient flows. Traditional payment systems could not compete with FinTech services in the context of sustainability, reliability, and attractiveness [5]. Over the past few years, a bundle of technological development paved the way for merchants to respond to customer needs for easy-to-use, resource-efficient, faster, and more secure payment methods. In retail stores and e-commerce, the usage of Fintech services such as mobile and contactless payment methods, Near-Field Communication (NFC)-enabled systems, peer-to-peer money transfers, one-click payment/checkout, micro-loans, and e-insurance has gradually become widespread. Significantly, the massive mobile phone and Internet access usage accelerate Fintech services. Recent digital transformation trends allow for innovative payment channels to be used in retail stores. Fintech providers allow the installation of digital payment methods on customers' mobile phones so that retailers and consumers gain maximum value.

Fintech transforms offline finance into a contactless, traceable, and sustainable mobile service. The proliferation of the everyday use of the Internet, smartphones, and mobile-based banking has changed customer shopping intentions. Based on information and communication technology (ICT), financial services improve and gain an innovative aspect. Fintech comprises big data, cloud technology, and social network services; it presents various services in response to customer needs, such as mobile payment, remittance, and crowdfunding. Thus, technology-enabled payment services push the limits of the traditional finance system and rapidly improve. Competitive financial attractiveness leads to the benefits of Fintech for users and merchants [6]. Using automated systems, Fintech companies release the profit to consumers operating costs while cutting unnecessary operating costs. Fintech usage provides a high frequency. If customers have free capital, investment in financial management and emergent capital requirements can be easily handled by Fintech applications. The useful construct of Fintech enables companies to collect and analyze large amounts of data resources. Fintech improvement offers digitalization and capitalization processes. The core technologies behind Fintech derive from strong IT expertise in the financial sector. Cybersecurity solutions, cloud technology, artificial intelligence, big data analysis, and distributed databases lead the new segment of Fintech companies.

Customers ask for green, reliable, faster, and convenient payment methods in the new technology-enabled era. Fintech services are the customer-centric strategy that enables merchants to present secure, faster, and innovative business models. In response to technology-oriented customer needs, merchants should integrate Fintech services into stores and e-commerce payment services. It is inevitable to adopt innovative technology in order to have a competitive advantage in the market and acquire more customer touchpoints. The selection and integration of convenient Fintech business models, design of entry, and correct timing are crucial for merchants [7,8]. While customer needs evolve, there must be a clear strategy for improving sustainability and remaining competitive. Mobile wallets are the crucial component of Fintech services that enables customers to shop via smartphones. The usage of mobile wallets has started to increase in stores and e-commerce, but the research in this field is limited. Because of the easy accessibility and significant importance to customers, wireless connection and Internet expansion make smartphones a critical channel for banking. M-banking takes advantage of the expanded use of smartphones and leads the demand for mobile wallet services among merchants and social customers [9,10]. Mobile wallet securely stores payment information, tokens, digital coupons, and loyalty card information and allows customers to purchase products in-store and online. This Fintech component integrates with banks, financial institutions, merchants, retailers, payment service providers, telecom operators, and many other businesses to present intelligent, practical, branded customer services [11].

Wadhera et al. [12] claim that there are four types of mobile wallets. Open wallets present cash withdrawal services in retailer or agent outlets; open wallets link with a bank and are reloadable (e.g., M-PESA). Semi-opened wallets link with a bank but do not allow cash withdrawal (e.g., Airtel Money). Semi-closed wallets allow customers buy products at merchants that integrate with the mobile wallet service provider. Semi-closed wallets are reloadable but inconvenient for redemption (e.g., Paytm). Closed wallets do not allow customers to withdraw cash; this type of wallet is non-reloadable with cash (e.g., gift vouchers). A mobile wallet is a digital wallet that benefits from different types of technologies. This service can include barcodes and Quick Response (QR) codes such as offered by Starbucks. A mobile wallet acts with QR technology as authentication for monetary transactions. Unmanned convenience stores such as Amazon's Go grocery store concept and Bingo Box offer automated services, including integrated mobile payment and business models. When consumers enter the stores, QR codes are simply scanned via smartphones. Consumers use the mobile wallet service at the end of the purchase process [13]. Also, mobile wallets comprise NFC technology and provide monetary transfers in short-range distances from one to another, such as in the Walmart NFC Payment System [14]. Customers can use proximity or contactless payment services through the NFC payment system. This is typically used for in-store or transportation services, where the customer completes the purchase by holding a smartphone on a reader [10].

NFC technology facilitates the payment process for consumers who can complete buying goods with the help of a cell phone in stores. NFC changes the payment process into a mere wave of the phone. NFC-based mobile wallet adoption is affected by perceived usefulness, perceived risk, trialability, compatibility, absorptive capacity, personal innovativeness in the new technologies, and attractiveness to alternatives [15]. Mobile wallet technology presents innovative, faster, and secure services while revealing a hosted solution for merchants, retailers, payment service providers, and financial institutions [10]. Integrated loyalty programs with mobile wallets provide ease of use in retail stores and online. Retailers rely on digital loyalty programs to enhance customer retention using mobile technology. This method lessens the burden on customers through the integration of mobile wallet services. Customers can be rid of carrying many loyalty cards [16]. Caro and Sadr [17] have posited that smart cards in the form of loyalty cards can be scanned at the point of sale (POS) or near the store entrance.

Mobile payment usage on the Web is a complex issue due to related factors such as the inclusion of multiple electronic payments and the secure exchange of payment information and receipts. Security and interoperability are guaranteed through emerging mobile wallets such as Apple Pay [18]. Chopdar et al. [19] have applied the Unified Theory of Acceptance and Use of Technology 2 (UTAUT2) to seek consumer adoption of mobile payment apps. According to the results, privacy risk and security risk impacts the consumer behavior. Internet and smartphone penetration stimulates a better customer experience with the help of mobile payment infrastructure. User-friendly features and omnichannel integration allow mobile wallets to present a customized shopping experience [20]. Also, the Internet of Things (IoT) option relies on the smartphones that customers carry. The wireless network in addition to a mobile payment method provides customer tracking during the store visit; this option supplies information about the overall customer experience [17].

Platform-based mobile payment service providers such as Facebook and Alibaba have strong IT expertise and mobile market understanding with subscribed mobile users. However, mobile payment providers lack consumer trust and financial market experience. IT expertise and integration ability are essential to retail partnerships. Price, mobility, convenience, self-expression, network externality, observability, personal information security, compatibility, trialability, social impact, quality, perceived risk, trust, and technical concerns are the crucial factors for mobile wallet integrations in stores and websites [21]. A mobile payment service requires a remarkable comprehension to enhance customer experience and reduce customer churn in online and offline environments such as Apple Pay, Samsung Pay, and Google Wallet. The most promising opportunity of mobile payment services from the

perspective of merchants is to reveal the maximum benefits and minimize risks. Complex authentication setups and confidentiality improve the awareness of security and privacy protection [22].

Over the last decade, mobile wallet services have evolved and attracted considerable attention from customers and companies. Considering the amount of $CO_2$ emitted by traditional checkout payment methods, mobile wallets that offer a paperless, reliable, and traceable payment service are also crucial in terms of sustainability. However, mobile wallets are a fairly new type of technology that is constantly evolving. According to the literature, there is not enough study in this field, especially before 2015, where the research was limited. Therefore, it is very difficult to find a literature review that focuses on mobile wallets. This situation creates the need to follow the development areas of mobile wallet technology to determine how to contribute more to sustainability and to direct future studies. In previous literature reviews, mobile wallets have been casually described as part of mobile payment trends or an emerging element of the Fintech ecosystem. Türkmen and Değerli [23] examined mobile wallets as a sub-topic of innovative banking trends in their literature review on how the financial sector has evolved over decades. Karsen et al. [24] and Dennehy and Sammon [25] conducted literature reviews on mobile payment. In these studies, mobile wallets are considered technological developments in mobile payments. The only study that we can say focuses on mobile wallets, albeit partially, was conducted by Ramli and Hamzah [26]. The study, which only examines mobile wallets from the adoption perspective basis of emerging economies in the literature, lacks in providing a holistic view. Due to the lack of a comprehensive mobile wallet literature survey, this article aims at filling this research gap by covering articles published between 2012 and 2022 over the Scopus, Web of Science, and Science Direct databases. Through this study, the adoption of technology for mobile wallets, as well as the customer engagement, competitiveness, and security and privacy issues, are grouped and detailed in the literature. In addition to the systematic literature review, we present a better understanding of mobile wallet services and propose new recommendations to enhance future research. The conducted literature review revealed that mobile wallets could be promoted in terms of traceability, usefulness, transaction speed, and security. According to the findings, environmental traceability could be improved to trace customer carbon footprint via smartphones per transaction. In the context of sustainability, the examined literature pointed out that resource-efficient and innovative aspects of mobile wallets are critical to eco-friendly shopping, customer acquisition, and competitive advantage. The following section includes the methodology of the literature review. Section 3 presents the results with descriptive statistics among the publication years. Section 4 analyzes the review results with critical remarks. Section 5 presents the discussion, including the most common findings and research agenda, and Section 6 presents the conclusion.

## 2. Methods

The survey target for this article originates from state-of-the-art mobile wallet technology usage. Figure 1 demonstrates the survey method of this article. Firstly, scientific databases hosting articles related to the research context have been clarified. The survey was conducted using the Scopus, Web of Science, and Science Direct databases. Publications were restricted with a clear filtering policy and must meet the requirements. Thus, each publication must be in an article format written in English, be published in a peer-reviewed academic journal, have the full text available, and be published between 2012 and 2022. The search scope was the 'topic' (article title, abstract, and keywords). Keyword selection was not limited to the mobile wallet. Comprehensive research shows that digital wallets and electronic wallets (e-wallets) are based on mobile devices that refer to the same technology in the literature as well [27–30]. It is for this reason that the keywords used for the search are 'mobile wallet', 'digital wallet', and 'e-wallet'.

After abstract reading and skimming, a total of 232 articles met the inclusion criteria. Finally, the remaining papers were exposed to whole article reading. As a research hotspot,

128 articles were related to mobile wallets in Scopus, Web of Science, and Science Direct databases. The filtered publications were analyzed in terms of three dimensions: the year of publication, published database, and research topics. Hence, the article classification was structured to better understand the research trends in the mobile wallet field.

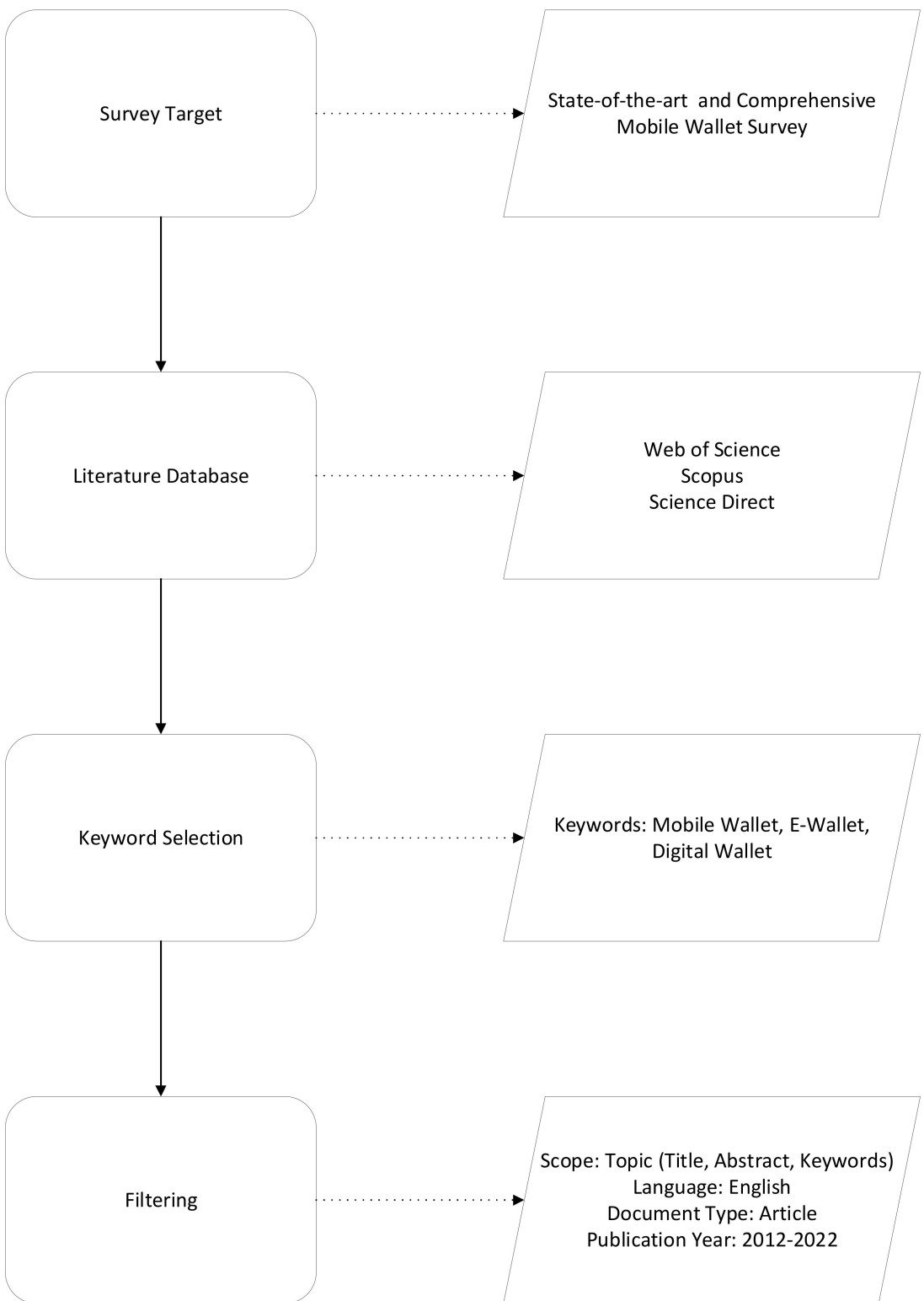

**Figure 1.** Survey method.

## 3. Results

### 3.1. Years of Publication

The publication years ranged between 2012 and 2022. As a new field of research, the term mobile wallet was rarely used before the last decade. Significantly, Fintech developments transformed mobile payment services, and adopting technology-oriented customers to a mobile wallet has increased the usage and research in this field. The number of publications has accelerated in the last few years. This status reflects the mobile wallet development and usage in stores and online channels. Figure 2 shows that this new research phenomenon has taken place after 2018.

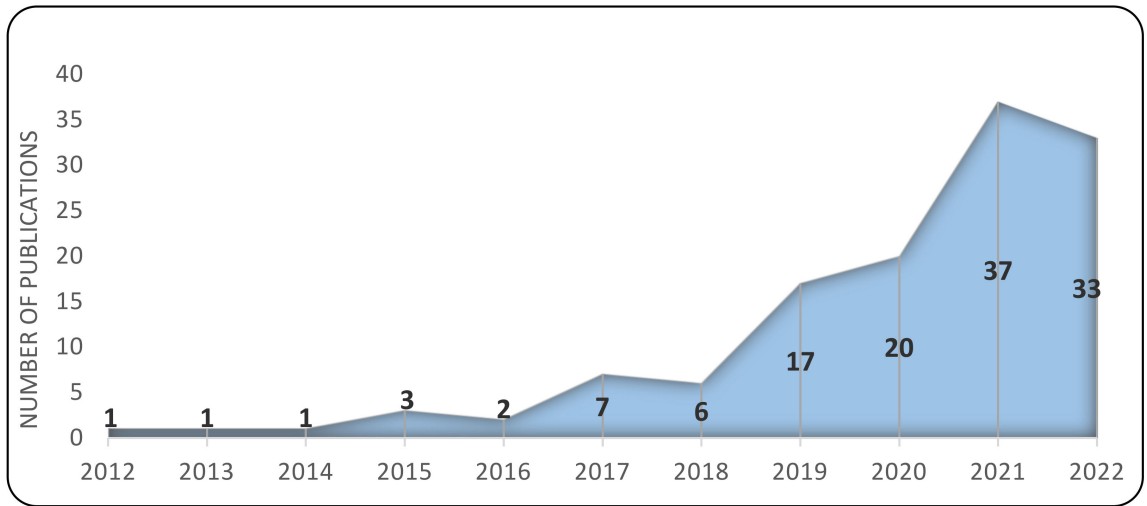

**Figure 2.** Number of publications by years. Source: the authors.

### 3.2. Academic Research Databases

Some of the most well-known academic research databases, namely Scopus, Web of Science, and Science Direct, were scanned during this study. The main objective of selecting several research databases was to present a comprehensive survey. Also, different databases cover the research diversity to better understand the academic contribution and direction. Afterward, we considered the criteria of proximity to sustainable, economic, managerial, and social issues related to mobile wallets to limit the selected articles during the entire article reading phase. Apart from these considerations, we have removed papers that directly focus on computer science and blockchain-based wallets for cryptocurrency. As a result, 128 filtered articles were analyzed in this study.

Figure 3 shows the database mapping of mobile wallets in the 128 articles obtained in the comprehensive literature review. Most articles are published in Scopus with a number of 97 articles, while the fewest number of articles are published in Science Direct, with 22 articles. The Web of Sciences database includes 58 articles in the 2012–2022 period. Additionally, nine of the articles are published in all three databases.

### 3.3. Research Topics

Observing how cutting-edge and eco-friendly services such as mobile wallets evolve to be and examining the development phases contribute to the understanding of how this service improves sustainability. After conducting a literature review on multiple aspects of mobile wallets, we discovered that mobile wallets are responsible for not only usefulness and enjoyment but also sustainability. When analyzing the different aspects of mobile wallets, we noted that this service evolves to the need of the customers and merchants. In the literature, it was possible to observe that the analyzed documents fit into three groups to understand how mobile wallets improve sustainability in payment services and analyze the research scope. Figure 4 illustrates the research topics in the period of 2012–2022. The

technology adoption of mobile wallets is the leading research topic in the literature. In the field of adoption, researchers seek to find an answer to research questions such as how customers increase to use of mobile wallets and which factors affect mobile wallet usage. Technology adoption includes determinants of intention to use a mobile wallet, pre- and post-acceptance dynamics, and factors influencing customer attitudes. Customer engagement is the second most prominent research topic, with 21 papers combining satisfaction, customer loyalty, and campaign/promotion management-related research for mobile wallets. Competitiveness is the third research topic, with 14 articles that comprise innovation and service quality-related research.

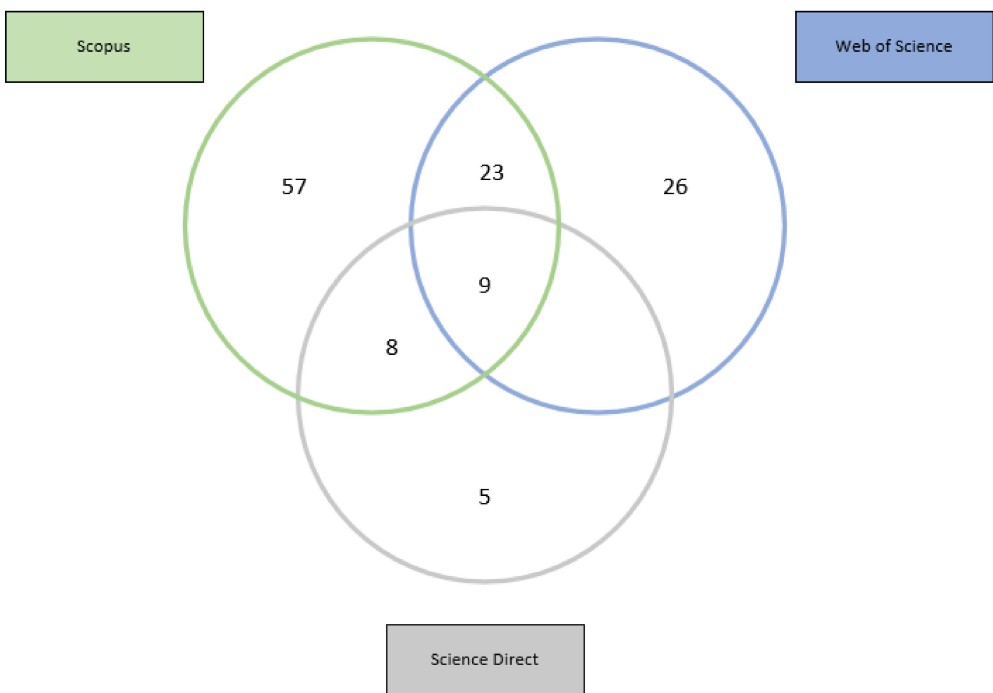

**Figure 3.** Academic research database mapping of the included mobile wallet articles.

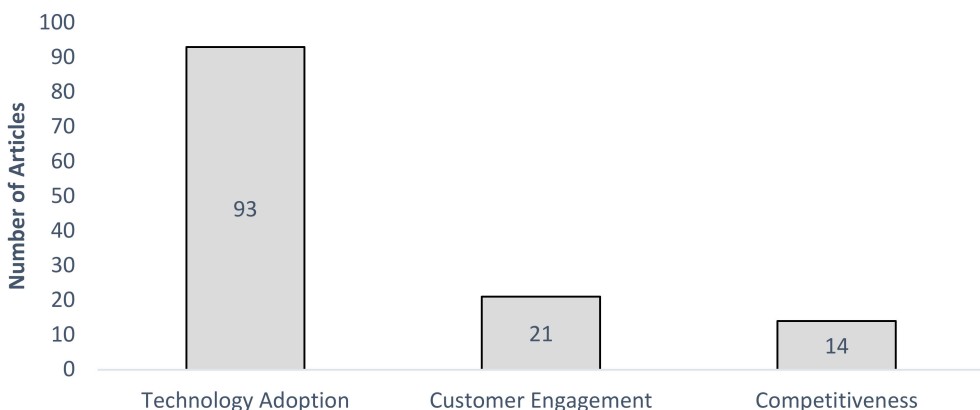

**Figure 4.** Research topics.

## 4. Analysis of Results

The comprehensive literature review in three databases showed that the research on mobile wallets is gathered into three main groups, namely technology adoption, customer engagement, and competitiveness. Furthermore, a summary literature table is presented in Appendix A.

*4.1. Technology Adoption*

A mobile wallet is a service that allows customers to sustainably make electronic transactions, fund transfers, and donations and use digital coupons and loyalty cards when shopping in-store and online through a smartphone application provided by a service provider [31,32]. Since the usage of this emerging technology starting from a decade ago, many researchers have sought to understand technology adoption criteria from the customer perspective. Investigating mobile wallet research touchpoints is vital to understanding the development direction.

4.1.1. Intention to Use of Customers and Merchants

Identifying constructs related to adoption in the previous studies, Malik and Sharma [33] conducted a weight analysis to present significant and non-significant factors for mobile wallet adoption. Vasantha and Sarika [34] observed the differences that demographic variables such as personal innovativeness and age cause in mobile wallet intention to use. The payment methods of consumers profoundly impact the frequency and quality of products purchased. Shekhar and Jaidev [35] revealed that personal innovativeness and perceived enjoyment are associated with perceived ease of use over mobile wallets. Ariguzo and White [36] examined the differences between mobile wallet adopters and non-adopters through M-PESA adoption in Kenya. Dauda and Lee [10] examined consumer preferences for future online banking services by conducting a conjoint analysis. The results revealed that Nigerian banks could strengthen competitive positioning with the aspect of entertainment and perceived ease of use of mobile wallets. Abbasi et al. [37] explored the continuance intention to buy of customers with the help of mobile wallets. They have employed a hybrid methodology to analyze customer survey data, namely structural equation modeling (SEM) and an asymmetrical analytical approach through fuzzy set qualitative comparative analysis (fsQCA). According to symmetrical outputs, service quality is the most critical factor in continuance intention via mobile wallets. However, asymmetrical outputs have cited that there needs to be a fusion of service quality, information quality, system quality, ease of use confirmation, usefulness confirmation, and security confirmation. User-related and system-specific factors contribute to NFC mobile wallet adoption [38]. Pham and Ho [15] have investigated mobile wallets as an NFC-based payment service. According to the findings, the most critical drivers of the adoption are the perceived usefulness and compatibility.

Shaw et al. [39] conducted a multinational study of mobile wallet adoption. A proposed diffusion of innovations (DoI) model was used to investigate mobile wallet adoption in Canada, Germany, and the United States. Security, privacy, and ubiquity were added to the model as context-relevant constructs. The results supported the proposed model while the ubiquity, privacy concerns, and security risks varied by country. Service providers should be able to ensure that mobile wallets' privacy and security policy complies with the user's perception [40]. Also, the reputation of mobile wallet service providers significantly impacts trust, perceived security, and continuance intention [41]. Nguyen and Nguyen [42] have suggested that to attract consumers to use mobile wallets, payment service providers should focus on the usefulness, ease of use, safety, and convenience of mobile wallet services. Phuong et al. [43] stated that mobile application quality, mobile wallet familiarity, situational normality, payment security, and the feedback mechanism are the mobile wallet features that influence customer intention to use a mobile wallet in Vietnam. According to the analysis, payment security and feedback mechanisms positively affect customer trust, satisfaction, and intention to use. However, mobile application quality and mobile wallet familiarity influence perceived ease of use, perceived usefulness, and intention to use mobile wallets. Talwar et al. [44] analyzed cross-sectional data of first-time mobile wallet users to test a two-step framework that included the pre-adoption and post-adoption factors. The findings revealed that service quality and information positively influence initial trust. Also, perceived usefulness and continuation intention have a positive relationship. Similarly, Gupta et al. [45] studied an extended expectation–confirmation model to

understand the pre-adoption importance of post-adoption satisfaction and continuance intention. They showed that pre-adoption performance expectancies have an essential impact on the consumption-driven confirmation, influencing satisfaction, post-adoption perceived usefulness, and perceived security. Obidat et al. [46] have also researched the issue of post-adoption. The study, completed in Jordan, revealed that perceived usefulness and perceived ease of use factors are more effective than the subjective norm for customers to continue using mobile wallets.

By predicting the mobile wallet resistance through the ANN model, Leong et al. [14] found that education, income, perceived novelty, usage, risk, value, and tradition significantly affect adoption. Based on the diffusion of innovation (DoI) theory, Kaur et al. [27] addressed the lack of and need to understand the intention to use mobile wallets. Researchers conducted an extensive cross-sectional survey of smartphone usage. The outcome of the study revealed that relative advantage, complexity, compatibility, and observability are significantly associated with customer intention to use mobile wallets. On the other hand, trialability had no association with customer intention to use or recommend mobile wallets. In the era of a cashless economy, sending and receiving payments is considered a milestone. Smartphone usage quickly accelerates, with readily available Internet access in urban and rural areas. Menon and Ramakrishnan [47] identified the drawbacks, challenges, and preferences of mobile wallet adoption. Shaw and Kesharwani [48] identified the moderating effect of smartphone addiction on mobile wallet adoption. Perceived ease of use, perceived usefulness, and subjective norms created the research construct to test the intention to use. In this context, they underlined the importance of communication with the right consumer through proper channels and communication with the appropriate age group. Amoroso et al. [49] conducted research among smartphone users in the Philippines to understand mobile wallet intention use. Reciprocity, switching costs, trust, loyalty, habit, and future repurchase constructed a model in this context. The findings revealed that switching cost is not high as adoption, but habit is the most vital mediator.

Merchant intention to use mobile wallet technology in stores was tested to understand the perception and adoption of mobile wallet services. Among Indian merchants, perceived customer value addition and perceived usefulness are the most critical factors in merchant intention to use mobile wallets [50]. Gupta et al. [51] investigated the factors affecting merchants' intention to adopt mobile wallets and created a comprehensive model combining perceived trust and price value with the help of the UTAUT and TTF models. The research has shown that the task-technology fit is the most important of all variables influencing the intention to use mobile wallets. Shaw [52] explored mobile wallet acceptance in Canada, where retailers are unwilling to invest in the technology to upgrade their store equipment until customers display a wider acceptance. The proposed model was built using the technology acceptance model (TAM) expanded with context relevant factors such as informal learning and trust. The findings revealed that mobile wallet acceptance is mainly related to perceived usefulness, whereas informal learning is mediated by trust. Tripathi et al. [53] proposed a model incorporating the attitude and subjective norms supporting the perceived usefulness and intention to use mobile wallets in small brick-and-mortar retailers. They stated that perceived trust positively impacts the intention to use mobile wallets, while perceived cost is a negative factor reducing intention to use.

### 4.1.2. Drivers and Barriers

Although new customers are quite knowledgeable about mobile wallets, Shah et al. [54] indicated that mobile wallet adoption remains limited due to safety concerns. Gupta [55] showed that the risk factor negatively affects mobile wallet usefulness, reasons to use, and the purpose to use for customers. Mombeuil and Uhde [56] investigated the continuous intention to use mobile wallet technology among foreign customers living in China. WeChat Pay users were tested in the context of relative convenience, relative advantages, perceived privacy, and perceived security. Each of the independent variables positively

influence the intention to use mobile wallets. Furthermore, compared with the traditional payment methods, WeChat Pay offers more security, privacy protection, and relatively more convenient and advantageous conditions. Mombeuil [57] searched Chinese mobile wallet users to present how relative convenience, relative advantage, perceived privacy, and perceived security influence the renewed adoption intention of mobile wallets. According to the results, relative advantage and perceived security are the best predictors of renewed adoption. Sharma et al. [31] developed an integrated hierarchical model to present the complex relationship among inhibitors to mobile wallet acceptance in Oman. Anxiety towards new technology, lack of awareness of mobile wallet benefits, lack of new technology skills, and complexity of new technology are the main inhibitors to the intention to use mobile wallets. Building a mixed research model, Chawla and Joshi [58] cited that perceived ease of use, perceived usefulness, security, trust, facilitating conditions, and lifestyle compatibility are the key factors explaining the intention to use mobile wallets among Indian users. Talwar et al. [59] focused on the positive and negative word of mouth (PWOM and NWOM, respectively) and examined the continued intention to use mobile wallets resulting from the WOM valence; it was proven that perceived information quality, perceived ability, and perceived benefit significantly affect PWOM, while perceived risk, perceived cost, and perceived uncertainty are the critical factors of NWOM. However, only PWOM spurs continuance intention to use a mobile wallet.

Singh et al. [60] explored the drivers related to customer intention, perceived satisfaction, and recommendations for using mobile wallet services in India. The study uncovered that ease of use, perceived risk, usefulness, and attitude are the most influential factors for explaining the intention to use, perceived satisfaction, and recommendation to use a mobile wallet. Furthermore, the studies revealed that the recent Know Your Customer (KYC) process of mobile wallets includes security concerns, low process efficiency, poor customer experience, and data protection issues. It is vital that customers can reach KYC-related documents and control personal data anytime in digital assets. Hassan and Shukur [61] claimed that the design of mobile wallet authentication on smartphones should be considered in terms of the intensity of security. Factors such as KYC verification attacks, SIM swapping, and app cloning affect the perceived security of mobile wallet users. Schlatt et al. [62] indicated that the blockchain-based infrastructure of mobile wallets enables customers to manage digital identity through creating and storing IDs and cryptographic keys. Moreover, it allows financial institutions to monitor suspicious customer behavior and detect fraud [63,64]. Kavitha et al. [65] tried to solve customers' security concerns using a queuing model. They benefited from the features of blockchain technology, which do not allow the change of records without notifying all participants and validating signatures.

The complexity of the global digital ecosystem generates new risks comprising cyberattacks and the threat of data misuse. Such privacy-related risks damage customer confidence and the reputation of the entity. Researchers have proposed various models to assess mobile wallet compliance with the General Data Protection Regulation (GDPR). The findings have revealed that country-level data privacy and protection practices are critical to evaluating country-level risk assessment [66,67]. Iqbal et al. [68] studied fingerprint verification technology in the case of the mobile wallet. According to the results, elderly people perceive biometric verification technology as a privacy risk and feel insecure about shopping. Adopting a design science approach, Akanfe et al. [69] analyzed the privacy policy of mobile wallet apps used in different countries to obtain the financial inclusion score.

Lew et al. [70] analyzed mobile wallet adoption in the hospitality industry. They employed self-efficacy, critical mass, and flow theories to clarify the intention to use a mobile wallet. Self-efficacy, perceived critical mass, mobile usefulness, mobile ease of use, mobile self-efficacy, and perceived enjoyment are the key inhibitors that have a significant effect on mobile wallet adoption. Leon [29] examined a Colombian mobile wallet network based on a dataset of daily bilateral transfers between users. Beyond the person-to-person transfers, customers frequently use mobile wallets for person-to-business and business-

to-business transfers. This increase in complexity proves the adoption of mobile wallets among Colombian users. Campbell and Singh [71] studied the behavioral intention to use mobile wallet service and customer innovativeness in digital payment adoption in India. Researchers claimed that perceived ease of use is a distinctive factor in the behavioral intention to use mobile wallets. Surprisingly, perceived usefulness and innovativeness do not positively affect the intention to use. In their study on how perceived regulatory support and promotional benefits affect mobile wallet adoption, Madan and Yadav [72] showed that performance expectancy, social influence, perceived risk, perceived value, facilitating conditions, and perceived regulatory support and promotional benefits are significant factors on the intention to use a mobile wallet. Searching for loyalty, satisfaction, and repurchasing intention of mobile wallet applications in Thailand, Amoroso and Ackaradejruangsri [73] discovered that personal innovativeness, perceived ease of use, and perceived usefulness are strongly correlated with customer attitudes. Senali et al. [74] found that personal innovativeness negatively moderates the effect of perceived compatibility on intention to adopt mobile wallets. Adjei et al. [75] verified that customer satisfaction influences mobile financial services and wallets' continual use. Chaddha et al. [76] investigated the reputation of celebrity endorsers on shopping intention toward mobile wallets in India. The outcome of the study confirmed that reputation constructs of the TEARS model positively influence purchasing intentions through a digital wallet. Various technology products provide an online customer experience at physical stores. Vidushi and Kashyap [77] designed a model showing that the change in purchase intention is due to mobile wallets, digital signage, smartphone, and click and collect from store technology.

Based on the UTAUT model, a proposed framework was applied to test trust in mobile wallet services and trust in mobile wallet service providers [78]. Estiyanti et al. [79] investigated the intention to use a mobile wallet in Indonesia. They proposed a model combining TAM with perceived usability. The results showed that mobile wallet adoption through perceived usability, usefulness, and enjoyment are positively effective factors. Using an extended TAM approach, To and Trinh [80] shaped the main factors explaining behavioral intention to use a mobile wallet in Vietnam. They found out that perceived enjoyment, perceived ease of use, and perceived usefulness positively impact the intention to use mobile wallets. George and Sunny [81] explored technology adoption models, the influence of various factors, and behavioral studies to understand continued intention to use and actual usage of mobile wallets. For this reason, they built a comprehensive conceptualization of mobile wallet adoption. Through an extended TAM, Seetharaman et al. [82] attempted to analyze key factors influencing mobile wallet acceptance in Singapore. The proposed model was created to understand the effects of innovativeness, critical mass, transaction security, transaction speed, trust, flexibility, cost of the transaction, availability of alternatives, consumer privacy, and anonymity on mobile wallet adoption. Singh et al. [83] found a significant association between perception, usage, satisfaction, and preference, whereas hedonism, security, and trust are less important to mobile wallet adoption. An integrated UTAUT model was proposed to understand mobile wallet intention to use. Chauhan et al. [84] cited Indian banks paying attention to the limited customer acceptance of Fintech-related services such as e-banking, mobile payment, and mobile wallet, which is not wider in society. Extending the UTAUT2 model with factors such as consumer innovativeness, perceived risk, and security information availability, they identified consumer intention to use various e-banking services. Alaeddin et al. [85] tested switching attitude and intention from a physical wallet to a mobile wallet. Based on the analysis, the perceived usefulness and perceived ease of use are the substantial factors in consumer attitude towards switching. Anshari et al. [86] expressed that pairing with the rising Internet connectivity and constant interaction with the technology leads to digital wallet adoption for the millennial generation. Using an extended TAM model, Soe [87] showed that perceived usefulness was positively affected by the stable Internet connection of mobile wallet users, and governments had a duty in this regard. Singh and Ghatak [88] used an extended TAM model with risk, compatibility, cost, usefulness, ease of use, be-

havioral intention to use, and actual usage to understand mobile wallet adoption. The theory of planned behavior (TPB) model was used to measure the behavioral intention to use the mobile wallet of Generation Z for in-store and online transactions [89]. Thanigan et al. [90] extended the UTAUT model via perceived value, perceived credibility, and technology anxiety factors. The authors compared the extended model to the original UTAUT model. As a result, regarding overall explanatory power, the extended UTAUT model was detected as robust. Conducting a meta-analysis, Bommer et al. [91] investigated the relationship between mobile wallet adoption and the intercorrelation of adoption antecedents obtained from an extended UTAUT model. Price value, social influence, hedonic motivation, and facilitating conditions show a positive impact on intention to use a mobile wallet. Bailey et al. [92] explored mobile payment adoption among Latin American customers by examining a bank-sponsored mobile wallet. For this purpose, they applied a revised UTAUT2 model, comprising performance expectancy, effort expectancy, bank trust, consumer innovativeness, consumer optimism, facilitating conditions, perceived quality, and consumer insecurity. Proposing an extended UTAUT model with perceived cost, perceived risk, and demonetization effect, Sobti [93] attempted to understand the antecedents of the behavioral intention to use mobile wallets and mobile banking in India. The research outcome verifies that demonetization and facilitating conditions impact technology adoption. Chawla and Joshi [94] have synthesized TAM and UTAUT models to set a research method. The authors suggested that mobile wallet service providers may focus on perceived usefulness, security, and lifestyle compatibility. Wamba et al. [95] integrated external factors, human orientation, social collectivism, and extended TAM constructs to observe the intention to use a mobile wallet in Cameroon. Sukwadi et al. [96] attempted to understand mobile wallet adoption in Indonesia with the help of an extended TAM. Considering the research output, the social influence, perceived ease of use, and mobility corroborated the intention to use mobile wallets. To research consumer-related variables affecting mobile wallet adoption, Amoroso and Magnier-Watanabe [11] proposed a comprehensive model to present the rapid diffusion of mobile wallets using the case of the Mobile Suica in Japan. Applying an extended UTAUT model, Limantara et al. [97] validated that performance expectancy, social influence, and perceived risk are essential for mobile wallet intention to use in Indonesia.

4.1.3. Differences in Perception of Technology

Lee et al. [98] studied satisfaction and perceived enjoyment of using the mobile wallet for Generation Y and Z in Malaysia. The findings revealed that the perceived enjoyment of shopping with a mobile wallet positively affects the buying impulse. At the same time, satisfaction has no direct effect on the buying impulse among Generation Y and Z. Sarmah et al. [99] analyzed millennials' intention to use mobile wallets. The outcome of the study showed that perceived ease of use and perceived usefulness explain the intention to use; however, trust has a significant impact on the actual use. Exploring mobile wallet intention to use among young Indian users, Kumar et al. [100] revealed that perceived usefulness, perceived ease of use, and perceived security significantly impact adoption and satisfaction. Taheam et al. [101] traced the factors driving mobile wallet usage among young people in India. They suggested that service providers consider controllability, security, social influence, perceived usefulness, and performance enhancement factors when designing mobile wallets.

The rapid increase in mobile wallet usage was discovered during the pandemic period. Alswaigh and Aloud [102] indicated that lifestyle compatibility, facilitating conditions, perceived usefulness, and perceived ease of use impressed customer behavior under the COVID-19 pandemic conditions. Meanwhile, a pandemic outbreak in Vietnam encouraged customers to adopt mobile wallet usage. Ly et al. [103] showed that trust, price-saving orientation, effort expectancy, and social influence lead to the intention to use. Al-Qudah et al. [104] conducted a study on the COVID-19 pandemic. The results revealed that skillfulness is the variable that most influences the intention to use mobile wallets,

followed by perceived usefulness and convenience of application. Ming and Jais [105] affirmed that perceived usefulness, perceived risk, government support, and social influence positively related to the attitude toward mobile wallet usage during the COVİD-19 pandemic. Astari et al. [106] showed that the moderating results of fear of the COVID-19 pandemic on attitudes increased intention to use a mobile wallet. Ojo et al. [107] confirmed the importance of perceived COVID-19 risk, perceived government support, and facilitating conditions on influencing mobile wallet intention use. Jaiswal et al. [108] searched for pre- and post-adoption factors of mobile wallet usage. The results confirmed that individual mobility, performance expectancy, effort expectancy, and facilitating conditions are the critical antecedents of intention to use. Thaker et al. [109] further identified performance expectancy, social influence, hedonic motivation, trust, facilitating condition, and habit as the main inhibitors of mobile wallet use in Malaysia. Lui et al. [110] examined Alipay to improve the performance of mobile wallet adoption in Malaysia. They validated that compatibility and perceived usefulness are the motivation behind the intention to use. Reddy and Rao [111] classified mobile wallet users to grasp the behavioral intention to use in different customer clusters. Tran and Hien [112] discovered that the perceived value of mobile wallets increases customer commitment and recommendation to use mobile wallets. Fanuel and Fajar [113] suggested a model for finding mobile wallet adoption drivers. They combined the TAM model with social environment and technological characteristics, thereby detecting the importance of personal experience, job relevance, perceived security, and subjective norms. Studying the moderating effect of gender and age between antecedents of mobile wallet adoption, Chawla and Joshi [114] have verified that more male and young customers have an intention to use mobile wallets. Reddy and Rao [115] have explored the moderating effect of gender among smartphone users in Spain. According to the findings, women are more influenced by personal innovativeness, while the social environment is more critical for men. While conducting research in India, Sharma et al. [116] explained the customer intention to use mobile wallets and its relationship with flexibility of usage, transaction speed, mobility, convenience, trust, usage cost, perceived ease of use, privacy, and anonymity. Malik et al. [117] ascertained that trust, incentive, and performance expectancy are positive factors. In contrast, social influence, enjoyment, aesthetics, and ease of use have no direct effect on mobile wallet app adoption. Kavitha and Kannan [118] took advantage of an extended TAM model to clarify that perceived usefulness, perceived ease of use, and perceived risk are vital factors for mobile wallet adoption.

*4.2. Customer Engagement*

Beyond the transaction, customer engagement is an ongoing relationship between the customer and the firm. In an emerging service such as the mobile wallet, it is vital to gain a competitive advantage through correct strategies to maintain customer engagement. Proposing a research model to observe mobile wallet integration with customer engagement, Kumar et al. [119] found that mobile wallet integration has an S-shaped relationship with customer engagement.

Exploring the customer experience of mobile wallets, Shankar and Behl [120] employed a data-driven mixed-method approach. Also, Ocak and Cagiltay [121] investigated customer experience through cognitive modeling and end-user usability testing on the touch screen of a mobile wallet app. The results showed that convenience, contact, interactivity, privacy, and security are the customer experience drivers.

Customer loyalty is a critical process for customer engagement, and users specifically decide whether or not to continue using mobile wallet brands. Gong et al. [122] examined mobile payment brand equity and customer loyalty using data from the AliPay and WeChat wallets. The findings revealed that platform application, application service, and service strategy complementarities positively impact brand equity and customer loyalty. Manickam et al. [123] investigated India's mobile wallet usage trend. According to the results, customers tend to use mobile wallet brands such as Google Pay and Paytm. Despite the loss of money, transaction failure, and network problems, customers stick to these two

brands because of their loyalty. Matemba et al. [124] studied customer loyalty factors on mobile wallet usage in the case of WeChat wallets. The authors found that convenience, social influence, and perceived availability of merchant support build customer loyalty. Chohan et al. [125] studied customer loyalty to QR code usage in digital payments and mobile wallets. According to the findings, customers' satisfaction, trust, and commitment to make payments with QR Codes increase customer loyalty. Aparna et al. [126] analyzed mobile wallet usage in India to build an efficient marketing strategy and build customer engagement. The results showed that customers use mobile wallets for recharging, booking transportation, and movie tickets. Transaction speed, attribute of style, and instant cashback are the reasons behind the higher customer engagement of mobile wallets [127]. Lo et al. [128] conducted a case study designing a mobile wallet for the needs of local people in Canada, aiming to understand strategies that engage with the customer better and roll out the product.

User awareness is a crucial factor in boosting mobile wallet service. In this context, Mathiraj et al. [129] evaluated consumer perception of the mobile wallet. They used the Hendry Garret Ranking method to identify mobile wallet service challenges, levels of perception, and satisfaction. Budiarani et al. [130] employed a Kano model to evaluate the service quality of mobile wallets widely used for online shopping. Satisfaction map findings showed that service providers must ensure customer satisfaction by improving items placed in the indifferent quadrant. Foster et al. [131] investigated the effect of product knowledge and risk perception on the satisfaction of mobile wallet users. The findings revealed a positive and significant effect. George and Sunny [132] have developed a comprehensive model, adding a promotional offer and situational influence of COVID-19 to the classic TAM model. The research results showed that the effects of satisfaction and the situational influence of COVID-19 on the continued usage intention of mobile wallets are strong. Okonkwo et al. [133] investigated the perception of mobile wallets in Cameroon, a cash-based economy. Analyses using SEM revealed that contactless financial transactions are not sufficiently compatible with the lifestyle of customers in cash-based economies.

Mobile wallet firms use social media for customer acquisition, relationship management, and promotion to increase popularity [134]. It is a substantial strategy that promotes offers to attract mobile wallet users [135]. Extracting social media sharings, Grover and Kar [136] have explored the customer dynamics and service advertisement of mobile wallets. The findings revealed that entertainment, remuneration, information, and social-based sharings are valuable for customer acquisition. Furthermore, firms need to focus on periodic campaigning and increasing network size. Lim et al. [137] examined the innovative function of money gifts over mobile wallet intention to use. The findings demonstrated that the gift functions promote various positive outcomes. Teng and Khong [138] have examined mobile wallet user behavior by applying text mining to social media platforms. According to the results, cashback, rewards, and promotional campaigns are attractive to customers. Aji and Adawiyah [139] have investigated how mobile wallets encourage spending among young customers. They suggested that promotions, perception of having more money, perceived easiness, and self-control affect young mobile wallet users.

*4.3. Competitiveness*

Service providers may focus on mobile wallet service quality to gain a competitive advantage over rivals. Kapoor et al. [140] indicated ten dimensions to measure service quality based on the literature. Applying a fuzzy TOPSIS approach, they prioritized the service quality drivers to enable companies to set marketing strategies. Kapoor et al. [141] suggested six critical dimensions of mobile wallet service quality: convenience, aesthetics, accessibility, security/reliability, responsiveness, and information quality. With the help of these service quality dimensions, they employed the fuzzy TOPSIS approach to rank mobile wallet alternatives in India. Ilankumaran [142] suggested that Fintech methods such as mobile wallets become more advantageous than traditional payments by eliminating the paper-clearing in banking transactions. Kumar et al. [143] used a time-series methodology

to sustain mobile wallet intention to analyze the COVID-19 pandemic period. According to the results, mobile wallet usage accelerated during the pandemic period. Policymakers need to enhance infrastructure, regulation, marketing strategies, and incentives to sustain mobile wallet service in the post-pandemic period.

Ease of use, instant money transfer without using cash, hedonic motivations, transaction security, and privacy cause mobile wallet usage to be more advantageous than traditional payment services. Customers exhibit the innovation-oriented and sustainability-oriented motivation to use a mobile wallet [144]. Al-Badi et al. [145] aimed to identify mobile wallet drivers and barriers in Oman. One of the critical studies for sustainability in mobile wallet usage showed that the commonly reported barriers are security, awareness, merchants, and financial support. Bagla and Sancheti [146] remarked that the challenges of mobile wallets are generated by the gaps between customer expectation and satisfaction. Rana et al. [147] investigated mobile wallet sustainability and highlighted two critical factors as implementation challenges. The lack of robust regulatory compliance and customer perception of the value of using mobile wallets are fundamental challenges for implementation. Kumar et al. [148] used the main path analysis of the network to describe technological trajectories in mobile payment technology. According to the results, mobile payment technology can be separated into mobile financial transaction systems, mobile wallet services, and payee mobile device payment selection systems. Firms may focus on each landscape to be more competitive in the market. This situation accelerates gaining a competitive advantage in a rapidly growing mobile wallet market. Alam et al. [149] cited that firms may determine strategies to leverage strengths and opportunities and overcome weaknesses and threats. They conducted an SWOT analysis to identify the challenges and prospects of using mobile wallets in Malaysia. The findings revealed that companies need to pay attention to the lack of infrastructure as weakness and security concerns, such as cyber-attack threats.

The growth of mobile wallets leads to cashless transactions and economic welfare. Technology advancement contributes to continued mobile wallet intention use [150,151]. Nurcahyo and Putra [152] employed the TOE framework, AHP, and TOPSIS methods to identify the inhibitors for collaborative decision-making between e-commerce and service providers. The research validated that a critical priority is the provision of mobile wallet payments. Networks and cooperation, management commitment, and expertise are the other critical factors to competitiveness in e-commerce service. Increasing Internet access and smartphone penetration provide enormous opportunities for the banking industry. Mobile wallet innovations increase Internet banking usage, and ATMs drive a wedge between banking institutions and the traditional form of brick-and-mortar branch banking [153]. Omarini [154] has investigated the mobile wallet ecosystem and potential sources of competitive advantage for retail banks. According to the research, mobile wallets are essential for banks to become more customer-centric.

The available literature has shown that most research has been on technology adoption using acceptance and behavioral models to understand customer shopping habits. Researchers develop ideas to facilitate the process by investigating how users think and face obstacles using mobile wallet technology.

## 5. Discussion

Fintech is a technology-enabled financial innovation that transforms traditional transaction methods into a mobile and eco-friendly system. It is a state-of-the-art technology that has come up in recent years related to Industry 4.0 developments. The massive utilization of mobile phones and Internet access significantly accelerates Fintech-based services in stores and online channels. Secure, green, faster, and convenient payment alternatives of Fintech services are the key factors to attracting technology-driven shoppers. Mobile wallets are one of the essential components of Fintech services that enables customers to shop through smartphones. Mobile wallets improve sustainability in payment services as it allows for cashless transactional methods, cost-efficient services, and traceable options.

Mobile wallets have evolved and drawn considerable attention from customers and companies in the context of emerging innovative payment methods in the last decade. In the competitive environment, mobile wallet technology enables firms to gain a competitive advantage and customer acquisition. Considering technology-driven and eco-friendly customer preferences, firms must respond to customer needs through online and in-store mobile wallet integrations. In this context, the secure, green, easy-to-use, and transaction speed features not only cause mobile wallet services to increase in popularity among customers, but also enable the wallets to offer a more sustainable service. However, research on mobile wallets in the literature is limited and has only begun to increase in the last few years. Due to the need for a comprehensive mobile wallet literature survey in the context of sustainability, this study aimed at filling this research gap by covering articles published between 2012 and 2022 over the Scopus, Web of Science, and Science Direct databases. In addition to conducting a systematic literature review, we presented a better understanding of mobile wallet services and sustainability contribution while proposing new recommendations to enhance future research. For this purpose, the literature was analyzed from the perspective of sustainability, economy, management, and social sciences. Articles that directly examined cryptocurrency trading and mechanism in the context of computer science were excluded. The literature review showed that the documents examined were divided into technology adoption, customer engagement, and competitiveness.

The available literature has shown that most research has been on technology adoption using acceptance and behavioral models to understand customer shopping habits. This situation may be considered as usual to understand the acceptance criteria for a reasonably new technology. Researchers develop ideas to facilitate the process by investigating how users think and face obstacles using mobile wallet technology. The most common findings pointed out that while perceived usefulness, enjoyment, and transaction speed positively affect mobile wallet use, factors such as perceived safety and privacy can often negatively affect the intention to use. However, some researchers have investigated the function of mobile wallet technology on customer engagement. With the service provided by mobile wallet technology in use, customer loyalty and customer acquisition issues has attracted attention. Research has shown that the underlying reason merchants invest in mobile wallet services in their stores and are enthusiastic to is to increase the perceived usefulness and customer lifetime value. The strategies developed by mobile wallet service providers and firms to gain a competitive advantage also led to research on competitiveness. The literature review has revealed that service quality and eco-friendly aspects of mobile wallets in particular positively contribute to competitive positioning. In addition, security and privacy in mobile wallets have drawn attention as topics that have begun to be researched. New methods have been started to strengthen wallet security, which customers can perceive as a weakness of mobile wallets compared with traditional payment services. The continuous development of mobile wallet services, the desire of different sectors to offer this technology, and the developed authentication mechanisms lead research in this direction.

Mobile wallet service provides sustainable, resource-efficient, and innovative solutions. In other words, mobile wallet improvements have significant value, and firms must develop strategies for sustainability, customer acquisition, and competitive advantage. Mobile wallets can be promoted in terms of traceability, usefulness, transaction speed, and security, but there are still many development fields and challenges to intention to use. The conducted literature review revealed potential research gaps as well. The research on technology adoption can be expected to continue as it is a relatively new and emerging technology. According to the examined literature and the knowledge of the authors, there is not any study that investigates the mediating effects of innovation, discomfort, optimism, transaction speed, and trust factors on mobile wallet usage together.

Furthermore, studies evaluating the use of mobile wallets with sustainability effects are relatively few in the literature. Considering that research in this area will both provide a competitive advantage to merchants and reduce the concerns of eco-friendly customers,

it can be expected that wallet features will be improved. In this context, the gap in the literature can be filled with the development of environmental traceability to enable customers to track their carbon footprints with smartphones per transaction. Environmental traceability could enable customers with sustainability concerns to take precautions and help raise awareness. Accordingly, mobile wallet services can enable providers to offer more eco-friendly features and merchants to demand more electronic payment methods for a competitive advantage.

Another research area needing improvement, as found in the literature review, is the set of studies focusing on customer lifetime value in mobile wallet usage. This area can present a critical research agenda focusing on churn prevention and customer engagement. Regarding the critical importance of customer retention, presenting user-friendly, fast, and reliable mobile wallets to customers could be essential. In addition, safety and privacy factors, which may be perceived as a weakness by customers, will be on the agenda of future studies to develop mobile wallet applications through innovative technology. Developing mobile wallets that use blockchain technology in various aspects and observing customers' shopping behavior in terms of security and privacy may be the subject of research interest in the near future. In this regard, the research agenda of mobile wallets are expected to focus on developments in sustainability, customer lifetime value, safety, and privacy factors. As a result, future work may increase the acceptance of mobile wallets.

## 6. Conclusions

Mobile wallets have been the subject of research and continuous development in the last decade. Unlike traditional payment services, mobile wallets offer cashless, eco-friendly, and enjoyable services via smartphones. The relatively new mobile wallet service constantly evolves in line with customer expectations and technological developments, attracting merchants and customers alike. This study conducted a comprehensive literature review to understand this emerging technology better and guide future research.

Considering the average 3.78 g of $CO_2$ emission per transaction produced when traditional checkout payments are used, and with this amount being multiplied by the number of transactions, it is noticed that mobile wallets are essential for sustainability. Mobile wallet services can help alleviate environmental pollution by being cashless, cardless, and paperless. Getting rid of paper money, coins, and plastic cards are easy-to-use benefits and much greener practices that reduce environmental threats. Moreover, smartphones enable customers to easily trace money in mobile wallets. Given these eco-friendly features, observing how cutting-edge technology such as mobile wallets develops and examining the development stages contributes to understanding how this service improves sustainability.

Literature review studies so far have mentioned mobile wallets as a sub-topic of payment services or have only focused on technology acceptance factors. However, mobile wallet studies have accelerated for several years. In future research, we foresee that mobile wallet developments will be more frequently handled with a focus on sustainability and that its contribution to the environment will be further increased with its traceability feature. In this context, our study is one of the pioneers among studies examining the development of mobile wallets with a focus on sustainability. This study's main contribution and originality is its comprehensive mobile wallet literature review, classification of studies according to the topics, and establishing the concept of sustainability at the center of the research.

As for the limitations of this study, the reviewed documents depend on the three well-known databases used. Therefore, essential documents from different databases may be omitted from the study. Another limitation of the study is the exclusion of articles that describe digital wallet mechanisms focused on cryptocurrency through a technical computer science method. These studies can be added at the discretion of the researcher.

Consequently, the cashless, eco-friendly, and traceable features of the mobile wallet service will be a valuable field of study for future improvements in sustainability, as has been the case for the last decade.

**Author Contributions:** Conceptualization, E.H. and Ö.V.; methodology, E.H.; data curation, Z.T.K.; writing—original draft preparation, E.H., Ö.V., Z.T.K., D.T. and C.A.; writing—review and editing, E.H., Ö.V., Z.T.K., D.T. and C.A.; supervision, E.H. All authors have read and agreed to the published version of the manuscript.

**Funding:** This research received no external funding.

**Institutional Review Board Statement:** Not applicable.

**Informed Consent Statement:** Not applicable.

**Data Availability Statement:** Not applicable.

**Conflicts of Interest:** The authors declare no conflict of interest.

## Appendix A

**Table A1.** Mobile wallet literature survey.

| N | Year | Authors | Article Title | Method |
|---|------|---------|---------------|--------|
| 1 | 2012 | Amoroso, D.L., Magnier-Watanabe, R. | Building a Research Model for Mobile Wallet Consumer Adoption: The Case of Mobile Suica in Japan | Case Study |
| 2 | 2013 | Ariguzo, G.C., White, D.S. | Exploring Demographic Differences in the Adoption of Mobile Money: M-PESA in Kenya | Survey |
| 3 | 2014 | Shaw, N. | The Mediating Influence of Trust in the Adoption of the Mobile Wallet | Survey, SEM |
| 4 | 2015 | Dauda, S. Y., Lee, J. | Technology Adoption: A Conjoint Analysis of Consumers' Preference on Future Online Banking Services | Survey, Conjoint Analysis |
| 5 | 2015 | Pham, T. T. T., Ho, J. C. | The Effects of Product-Related, Personal-Related Factors and Attractiveness of Alternatives on Consumer Adoption of NFC-Based Mobile Payments | Survey, SEM |
| 6 | 2015 | Boro, K. | Prospects and Challenges of Technological Innovation in Banking Industry of North East India | Interview |
| 7 | 2016 | Madan, K., Yadav, R. | Behavioural Intention to Adopt Mobile Wallet: A Developing Country Perspective | Survey, SEM |
| 8 | 2016 | Taheam, K., Sharma, R., Goswami, S. | Drivers of Digital Wallet Usage: Implications for Leveraging Digital Marketing | Survey, SEM |
| 9 | 2017 | Campbell, D., Singh, C.B. | A Study of Customer Innovativeness for the Mobile Wallet Acceptance in Rajasthan | Survey, SEM |
| 10 | 2017 | Seetharaman, A., Kumar, K. N., Palaniappan, S., Weber, G. | Factors Influencing Behavioural Intention to Use the Mobile Wallet in Singapore | Survey, CFA |
| 11 | 2017 | Singh, N., Srivastava, S., Sinha, N. | Consumer Preference and Satisfaction of M-wallets: a Study on North Indian Consumers | Survey, Correlation and Regression Analysis |
| 12 | 2017 | Amoroso, D., Ackaradejruangsri, P. | How Consumer Attitudes Improve Repurchase Intention | Survey, SEM |
| 13 | 2017 | Shah, B., Ullatil, D.S., Nagendra, A. | Analysis of the Inception, Acceptance and Future of E-Wallets | Survey, Correlation and Regression Analysis |
| 14 | 2017 | Campbell, D., Singh, C.B. | A Study of Customer Innovativeness for the Mobile Wallet Acceptance in Rajasthan | Survey, SEM |
| 15 | 2017 | Ocak, N., Cagiltay, K. | Comparison of Cognitive Modeling and User Performance Analysis for Touch Screen Mobile Interface Design | Statistical Analysis, Video Analysis |

**Table A1.** *Cont.*

| N | Year | Authors | Article Title | Method |
|---|---|---|---|---|
| 16 | 2018 | Sharma, S. K., Mangla, S. K., Luthra, S., Al-Salti, Z. | Mobile Wallet Inhibitors: Developing a Comprehensive Theory Using an Integrated Model | Interpretive Structural Modelling (ISM), fuzzy MICMAC |
| 17 | 2018 | Kumar, A., Adlakaha, A., Mukherjee, K. | The Effect of Perceived Security and Grievance Redressal on Continuance Intention to Use M-wallets in a Developing Country | Survey, SEM |
| 18 | 2018 | Alaeddin, O., Rana, A., Zainudin, Z., Kamarudin, F. | From Physical to Digital: Investigating Consumer Behaviour of Switching to Mobile Wallet | Survey, SEM |
| 19 | 2018 | Matemba, E. D., Li, G., Maiseli, B. J. | Consumers' Stickiness to Mobile Payment Applications: An Empirical Study of WeChat Wallet | Survey, SEM |
| 20 | 2018 | Bagla, R.K., Sancheti, V. | Gaps in Customer Satisfaction with Digital Wallets: Challenge for Sustainability | Survey, Inferential Analysis |
| 21 | 2018 | Omarini, A.E. | Fintech and the Future of the Payment Landscape: The Mobile Wallet Ecosystem—A Challenge for Retail Banks? | Case Study |
| 22 | 2019 | Phutela, N., Altekar, S. | Mobile Wallets in India: A Framework for Consumer Adoption | Survey |
| 23 | 2019 | Shaw, B., Kesharwani, A. | Moderating Effect of Smartphone Addiction on Mobile Wallet Payment Adoption | Survey, SEM, Multi-group Analysis |
| 24 | 2019 | Chawla, D., Joshi, H. | Consumer Attitude and Intention to Adopt Mobile Wallet in India—An Empirical Study | Survey, SEM |
| 25 | 2019 | Sobti, N. | Impact of Demonetization on Diffusion of Mobile Payment Service in India Antecedents of Behavioral Intention and Adoption Using Extended UTAUT Model | Survey, SEM |
| 26 | 2019 | Sharma, D., Aggarwal, D., Gupta, A. | A Study of Consumer Perception Towards Mwallets | Survey, Interview, Regression Analysis |
| 27 | 2019 | Reddy, T.T., Rao, B.M. | The Moderating Effect of Gender on Continuance Intention Toward Mobile Wallet Services in India | Survey, SEM |
| 28 | 2019 | Malik, A., Suresh, S., Sharma, S. | An Empirical Study of Factors Influencing Consumers' Attitude Towards Adoption of Wallet Apps | Survey, Correlation and Regression Analysis |
| 29 | 2019 | Vasantha, S., Sarika, P. | Empirical Analysis of Demographic Factors Affecting Intention to Use Mobile Wallet | Comparison Analysis |
| 30 | 2019 | Menon, M.M., Ramakrishnan, H.S. | Revolution of E-wallets Usage among Indian Millennial | Survey, SEM |
| 31 | 2019 | Kumar, V., Nim, N., Sharma, A. | Driving Growth of Mwallets in Emerging Markets: a Retailer's Perspective | Qualitative Study |
| 32 | 2019 | Semerikova, E. | Payment Instruments Choice of Russian Consumers: Reasons and Pain Points | Exploratory Study |
| 33 | 2019 | Sarika, P., Vasantha, S. | Impact of Mobile Wallets on Cashless Transaction | Survey |
| 34 | 2019 | Mathiraj, S.P., Geeta, S.D.T., Saroja Devi, R. | Consumer Acuity on Select Digital Wallets | Survey, ANOVA, Hendry Garret Ranking Method |
| 35 | 2019 | Aparna, H., Karthika, S., Rajalakshmi, V.R. | A Study on the Digital Wallet Usage among Citizens of Kochi using FP-growth Algorithm | Survey, FP-Growth Algorithm |
| 36 | 2019 | Ilankumaran, G. | Payment System Indicators of Digital Banking Ecosystem in India | Trend Analysis |

**Table A1.** *Cont.*

| N | Year | Authors | Article Title | Method |
|---|---|---|---|---|
| 37 | 2019 | Nair, A.K.S., Bhattacharyya, S.S. | Is Sustainability a Motive to Buy? An Exploratory Study in the Context of Mobile Applications Channel among Young Indian Consumers | Survey, CFA |
| 38 | 2019 | David, S., Kathrine, J.W. | An Investigative Report on Encryption Based Security Mechanisms for E-Wallets | Review |
| 39 | 2020 | Leong, L. Y., Hew, T. S., Ooi, K. B., Wei, J. | Predicting Mobile Wallet Resistance: A Two-Staged Structural Equation Modeling-Artificial Neural Network Approach | Survey, SEM, ANN |
| 40 | 2020 | Talwar, S., Dhir, A., Khalil, A., Mohan, G., Islam, A. N. | Point of Adoption And Beyond. Initial trust and Mobile-Payment Continuation Intention | Survey, SEM |
| 41 | 2020 | Gupta, A., Yousaf, A., Mishra, A. | How Pre-Adoption Expectancies Shape Post-Adoption Continuance Intentions: An Extended Expectation-Confirmation Model | Survey, SEM |
| 42 | 2020 | Kaur, P., Dhir, A., Bodhi, R., Singh, T., Almotairi, M. | Why do People Use and Recommend M-Wallets? | Survey, SEM |
| 43 | 2020 | Kavitha, K., Kannan, D. | Factors Influencing Consumers Attitude towards Mobile Payment Applications | Survey, SEM |
| 44 | 2020 | Chawla, D., Joshi, H. | The Moderating Role Of Gender and Age in the Adoption Of Mobile Wallet | Survey, MGA |
| 45 | 2020 | Adjei, J.K., Odei-Appiah, S., Tobbin, P.E. | Explaining the Determinants of Continual Use of Mobile Financial Services | Survey, SEM |
| 46 | 2020 | Singh, N., Sinha, N. | How Perceived Trust Mediates Merchant's Intention to use a Mobile Wallet Technology | Survey, SEM |
| 47 | 2020 | Lew, S., Tan, G. W. H., Loh, X. M., Hew, J. J., Ooi, K. B. | The Disruptive Mobile Wallet in the Hospitality Industry: An Extended Mobile Technology Acceptance Model | Survey, SEM |
| 48 | 2020 | Mombeuil, C. | An Exploratory Investigation of Factors Affecting and Best Predicting the Renewed Adoption of Mobile Wallets | Survey, Hierarchical Ordinary Least Squares (OLS) Regression |
| 49 | 2020 | Phuong, N. N. D., Luan, L. T., Dong, V. V., Khanh, N. L. N. | Examining Customers' Continuance Intentions towards E-wallet Usage: The Emergence of Mobile Payment Acceptance in Vietnam | Survey, SEM |
| 50 | 2020 | Soodan, V., Rana, A. | Modeling Customers' Intention to Use E-Wallet in a Developing Nation: Extending UTAUT2 With Security, Privacy and Savings | Survey |
| 51 | 2020 | Iqbal, S., Irfan, M., Ahsan, K., Hussain, M. A., Awais, M., Shiraz, M., Hamdi, M., Alghamdi, A. | A Novel Mobile Wallet Model for Elderly Using Fingerprint as Authentication Factor | Survey, Association Analysis |
| 52 | 2020 | Grover, P., Kar, A. K. | User Engagement for Mobile Payment Service Providers—Introducing the Social Media Engagement Model | Content Analysis, Geospatial Analysis |
| 53 | 2020 | Gong, X., Cheung, C. M., Zhang, K. Z., Chen, C., Lee, M. K. | Cross-Side Network Effects, Brand Equity, and Consumer Loyalty: Evidence from Mobile Payment Market | Survey, SEM |
| 54 | 2020 | Singh, N., Sinha, N., Liébana-Cabanillas, F. J. | Determining Factors in the Adoption and Recommendation of Mobile Wallet Services in India: Analysis of the Effect of Innovativeness, Stress to Use and Social Influence | Survey, SEM |

**Table A1.** *Cont.*

| N | Year | Authors | Article Title | Method |
|---|---|---|---|---|
| 55 | 2020 | Hoang, H., Le, T.T. | The Role of Promotion in Mobile Wallet Adoption—A Research in Vietnam | Survey, SEM |
| 56 | 2020 | Akanfe, O., Valecha, R., Rao, H.R. | Design of a Compliance Index for Privacy Policies: A Study of Mobile Wallet and Remittance Services | NLP, LDA |
| 57 | 2020 | Kapoor, A., Sindwani, R., Goel, M. | Mobile Wallets: Theoretical and Empirical Analysis | Fuzzy TOPSIS |
| 58 | 2020 | Akanfe, O., Valecha, R., Rao, H. R. | Assessing Country-Level Privacy Risk for Digital Payment Systems | Privacy Policy Analysis |
| 59 | 2021 | Talwar, M., Talwar, S., Kaur, P., Islam, A. N., Dhir, A. | Positive and Negative Word Of Mouth (WOM) are not Necessarily Opposites: A Reappraisal Using the Dual Factor Theory | Survey, SEM |
| 60 | 2021 | Mombeuil, C., Uhde, H. | Relative Convenience, Relative Advantage, Perceived Security, Perceived Privacy, and Continuous Use Intention of China's WeChat Pay: A Mixed-Method Two-Phase Design Study | Survey, Hierarchical Ordinary Least Squares (OLS) Regression |
| 61 | 2021 | León, C. | The Adoption of a Mobile Payment System: The User Perspective | Network Analysis |
| 62 | 2021 | Estiyanti, N. M., Agustia, D., Mulia, R. A., Alfarisyi, R., Frandha, R., Hidayanto, A. N., Kurnia, S. | The Impact of Perceived Usability on Mobile Wallet Acceptance: A Case of Gopay Indonesia | Survey, SEM |
| 63 | 2021 | George, A., Sunny, P. | Developing a Research Model for Mobile Wallet Adoption and Usage | Hypothesis Testing |
| 64 | 2021 | Sarmah, R., Dhiman, N., Kanojia, H. | Understanding Intentions and Actual Use of Mobile Wallets By Millennial: an Extended TAM Model Perspective | Survey, SEM |
| 65 | 2021 | To, A.T., Trinh, T.H.M. | Understanding Behavioral Intention to Use Mobile Wallets in Vietnam: Extending the Tam Model with Trust and Enjoyment | Survey, SEM |
| 66 | 2021 | Singh, S., Ghatak, S. | Investigating E-Wallet Adoption in India: Extending the TAM Model | Survey, SEM |
| 67 | 2021 | Thanigan, J., Reddy, S. N., Sethuraman, P., Rajesh, J. I. | Understanding Consumer Acceptance of M-Wallet Apps: The Role of Perceived Value, Perceived Credibility, and Technology Anxiety | Survey, SEM |
| 68 | 2021 | Amoroso, D., Lim, R., Roman, F.L. | The Effect of Reciprocity on Mobile Wallet Intention: A Study of Filipino Consumers | Survey, SEM |
| 69 | 2021 | Persada, S. F., Dalimunte, I., Nadlifatin, R., Miraja, B. A., Redi, A. A. N. P., Prasetyo, Y. T., Chin, J., Lin, S. | Revealing the Behavior Intention of Tech-savvy Generation Z to Use Electronic Wallet Usage: A Theory of Planned Behavior Based Measurement | Survey, SEM |
| 70 | 2021 | Anshari, M., Arine, M. A., Nurhidayah, N., Aziyah, H., Salleh, M. H. A. | Factors Influencing Individual in Adopting Ewallet | Survey, Correlation and Regression Analysis |
| 71 | 2021 | Alswaigh, NY., Aloud, M.E. | Factors Affecting User Adoption of E-Payment Services Available in Mobile Wallets in Saudi Arabia | Survey, Correlation and Regression Analysis |
| 72 | 2021 | Chawla, D., Joshi, H. | Importance-Performance Map Analysis to Enhance the Performance of Attitude Towards Mobile Wallet Adoption among Indian Consumer Segments | Survey, SEM |

**Table A1.** *Cont.*

| N | Year | Authors | Article Title | Method |
|---|------|---------|---------------|--------|
| 73 | 2021 | Wamba, S.F., Queiroz, M.M., Blome, C., Sivarajah, U. | Fostering Financial Inclusion in a Developing Country: Predicting User Acceptance of Mobile Wallets in Cameroon | Survey, SEM |
| 74 | 2021 | Hor, H. L., Wong, W. L., Ho, S. K., Tan, J. H., Teo, S. X., Foo, P. Y. | The Leading Edge of NFC Mobile Wallet Adoption: An Empirical Analysis from an Emerging Economy's Perspective | Survey, SEM |
| 75 | 2021 | Reddy, T.T., Rao, B.M. | Determinants of Continuance Intention to Use Mobile Wallet Services: Light Users vs Heavy Users | Survey, Multivariate Data Analysis Techniques |
| 76 | 2021 | Tran Le Na, N., Hien, N. N. | A Study of User's M-Wallet Usage Behavior: The Role of Long-Term Orientation and Perceived Value | Survey, SEM |
| 77 | 2021 | Garrouch, K. | Does the Reputation of the Provider Matter? A Model Explaining the Continuance Intention of Mobile Wallet Applications | Survey, SEM |
| 78 | 2021 | Malik, A., Sharma, S. | Antecedents of Wallet App Adoption | Weight Analysis |
| 79 | 2021 | Limantara, N., Jovandy, J., Wardhana, A.K., Steven, Jingga, F. | Evaluation of One of Leading Indonesia's Digital Wallet Using the Unified Theory of Acceptance and Use of Technology | Survey, SEM |
| 80 | 2021 | Vidushi, V., Kashyap, R. | Reconfigure the Apparel Retail Stores with Interactive Technologies | Survey, SEM |
| 81 | 2021 | Chaddha, P., Agarwal, B., Zareen, A. | Investigating the Effect of the Credibility of Celebrity Endorsement on the Intent of Consumers to Buy Digital Wallets in India | Survey, SEM |
| 82 | 2021 | Fanuel, P.N., Fajar, A.N. | Digital Wallet War in Asia: Finding the Drivers of Digital Wallet Adoption | Survey |
| 83 | 2021 | Lui, T.K., Zainuldin, M.H., Yii, K.J., Lau, L.S., Go, Y.H. | Consumer Adoption of Alipay in Malaysia: The Mediation Effect of Perceived Ease of Use and Perceived Usefulness | Survey, SEM |
| 84 | 2021 | Aji, H.M., Adawiyah, W.R. | How E-Wallets Encourage Excessive Spending Behavior among Young Adult Consumers? | Survey, SEM |
| 85 | 2021 | Lo, K., Liu, F., Huang, J. | OneFeather Mobile Wallet: A Digital Solution for Indigenous Peoples in Canada? | Case Study |
| 86 | 2021 | Budiarani, V.H., Maulidan, R., Setianto, D.P., Widayanti, I. | The Kano Model: How the Pandemic Influences Customer Satisfaction with Digital Wallet Services in Indonesia | Kano Model, Survey, Exploratory Factor Analysis (EFA), CFA |
| 87 | 2021 | Kapoor, A., Sindwani, R., Goel, M. | Ranking Mobile Wallet Service Providers Using Fuzzy Multi-Criteria Decision-Making Approach | Fuzzy TOPSIS |
| 88 | 2021 | Nurcahyo, R., Putra, P.A. | Critical Factors in Indonesia's E-Commerce Collaboration | AHP, TOPSIS |
| 89 | 2021 | Alam, M. M., Awawdeh, A. E., Muhamad, A. I. B. | Using E-Wallet for Business Process Development: Challenges and Prospects in Malaysia | SWOT Analysis |
| 90 | 2021 | Kumar, V., Lai, K. K., Chang, Y. H., Bhatt, P. C., Su, F. P. | A Structural Analysis Approach to Identify Technology Innovation and Evolution Path: A Case of M-Payment Technology Ecosystem | Network Establishment, Main Path Analysis |
| 91 | 2021 | Schlatt, V., Sedlmeir, J., Feulner, S., Urbach, N. | Designing a Framework for Digital KYC Processes Built on Blockchain-Based Self-Sovereign Identity | Applied DSR Process |
| 92 | 2021 | Akanfe, O., Valecha, R., Rao, H.R. | Design of an Inclusive Financial Privacy Index (INF-PIE): A Financial Privacy and Digital Financial Inclusion Perspective | LDA, PCA |

**Table A1.** *Cont.*

| N | Year | Authors | Article Title | Method |
|---|------|---------|---------------|--------|
| 93 | 2021 | Hassan, M. A., Shukur, Z. | Device Identity-Based User Authentication on Electronic Payment System for Secure E-Wallet Apps. | Simulation |
| 94 | 2021 | Teng, S., Khong, K. W. | Examining Actual Consumer Usage of E-wallet: A Case Study of Big Data Analytics | Clustering, SEM |
| 95 | 2021 | Shankar, A., Behl, A. | How to Enhance Consumer Experience over Mobile Wallet: a Data-Driven Approach | Survey, Text Mining |
| 96 | 2022 | Abbasi, G.A., Sandran, T., Ganesan, Y., Iranmanesh, M. | Go Cashless! Determinants of Continuance Intention to Use E-wallet apps: A Hybrid Approach Using PLS-SEM and fsQCA | SEM, Fuzzy Set Qualitative Comparative Analysis (fsQCA) |
| 97 | 2022 | Shaw, N., Eschenbrenner, B., Brand, B. M. | Towards a Mobile App Diffusion of Innovations model: A Multinational Study of Mobile Wallet Adoption | Survey, SEM |
| 98 | 2022 | Thaker, H.M.T., Subramaniam, N.R., Qoyum, A., Hussain, H.I. | Cashless Society, E-Wallets and Continuous Adoption | Survey, SEM |
| 99 | 2022 | Chauhan, V., Yadav, R., Choudhary, V. | Adoption of Electronic Banking Services in India: an Extension of UTAUT2 model | Survey, SEM |
| 100 | 2022 | Tripathi, S.N., Srivastava, S., Vishnani, S. | Mobile Wallets: Achieving Intention to Recommend by Brick and Mortar Retailers | Reliability, Validity, and Mediation Analyses |
| 101 | 2022 | Lee, Y.Y., Gan, C.L., Liew, T.W. | Do E-wallets Trigger Impulse Purchases? An Analysis of Malaysian Gen-Y and Gen-Z Consumers | Survey, SEM |
| 102 | 2022 | Jaiswal, D., Kaushal, V., Mohan, A., Thaichon, P. | Mobile Wallets Adoption: Pre- and Post-Adoption Dynamics of Mobile Wallets Usage | Survey, Moderation and Multi-Group Analysis |
| 103 | 2022 | Ly, H.T.N., Khuong, N.V., Son, T.H. | Determinants Affect Mobile Wallet Continuous Usage in COVID 19 Pandemic: Evidence from Vietnam | Survey, SEM |
| 104 | 2022 | Sukwadi, R., Caroline, L.S., Chen, G.Y.H. | Extended Technology Acceptance Model for Indonesian Mobile Wallet: Structural Equation Modeling Approach | Survey, SEM |
| 105 | 2022 | Kumar, R., Ratra, V., Mandava, S. | Mobile Wallet Payments in the Time of COVID-19: The Indian Experience | Time-Series Technique |
| 106 | 2022 | Muhtasim, D.A., Tan, S.Y., Hassan, M.A., Pavel, M.I., Susmit, S. | Customer Satisfaction with Digital Wallet Services: An Analysis of Security Factors | Survey, Correlation and Regression Analysis |
| 107 | 2022 | Obidat, A., Almahameed, M., Alalwan, M. | An Empirical Examination of Factors Affecting the Post-Adoption Stage of Mobile Wallets by Consumers: A Perspective from a Developing Country | Survey, SEM |
| 108 | 2022 | Okonkwo, C.W., Amusa, L.B., Twinomurinzi, H., Fosso Wamba, S. | Mobile Wallets in Cash-Based Economies during COVID-19 | Survey, SEM |
| 109 | 2022 | George, A., Sunny, P. | Why do People Continue Using Mobile Wallets? An Empirical Analysis Amid COVID-19 pandemic | Survey, SEM |
| 110 | 2022 | Gupta, R.K. | Adoption of Mobile Wallet Services: An Empirical Analysis | Survey, Regression Analysis, SEM |
| 111 | 2022 | Gupta, S., Dhingra, S., Tanwar, S., Aggarwal, R. | What Explains the Adoption of Mobile Wallets? A Study from Merchants' Perspectives | Survey, SEM |

**Table A1.** *Cont.*

| N | Year | Authors | Article Title | Method |
|---|------|---------|---------------|--------|
| 112 | 2022 | Rana, N.P., Luthra, S., Rao, H.R. | Assessing Challenges to the Mobile Wallet Usage in India: An Interpretive Structural Modelling Approach | Interpretive Structural Modelling (ISM) |
| 113 | 2022 | Kavitha, R., Rajeswari, R., Mukherjee, P., Rout, S., Patra, S.S. | Performance Measures of Blockchain-Based Mobile Wallet Using Queueing Model | Queueing Model |
| 114 | 2022 | Al-Badi, A.H., Govindaluri, S.M., Sharma, S.K., Khan, A.I. | Global and Local Perspective on the Usage of Mobile Wallet | Interview, Statistics |
| 115 | 2022 | Manickam, T., Vinayagamoorthi, G., Gopalakrishnan, S., Sudha, M., Mathiraj, S.P. | Customer Inclination on Mobile Wallets with Reference to Google-Pay and PayTM in Bengaluru City | Survey, CFA |
| 116 | 2022 | Chohan, F., Aras, M., Indra, R., Wicaksono, A., Winardi, F. | Building Customer Loyalty in Digital Transaction Using QR Code: Quick Response Code Indonesian Standard (QRIS) | Survey, Statistics |
| 117 | 2022 | Astari, A.A.E., Yasa, N.N.K., Sukaatmadja, I.P.G., Giantari, I.G.A.K. | Integration of Technology Acceptance Model (TAM) and Theory of Planned Behavior (TPB): An E-Wallet Behavior with Fear of COVID-19 as a Moderator Variable | Survey, SEM |
| 118 | 2022 | Foster, B., Hurriyati, R., Johansyah, M.D. | The Effect of Product Knowledge, Perceived Benefits, and Perceptions of Risk on Indonesian Student Decisions to Use E-Wallets for Warunk Upnormal | Survey, SEM |
| 119 | 2022 | Senali, M. G., Iranmanesh, M., Ismail, F. N., Rahim, N. F. A., Khoshkam, M., Mirzaei, M. | Determinants of Intention to Use e-Wallet: Personal Innovativeness and Propensity to Trust as Moderators | Survey, SEM |
| 120 | 2022 | Nguyen, H.T., Nguyen, N.T. | Identifying the Factors Affecting the Consumer Behavior in Switching to e-wallets in Payment Activities | Survey, Correlation and Regression Analysis |
| 121 | 2022 | Soe, M.H. | Do They Really Intend to Adopt E-Wallet? Prevalence Estimates for Government Support and Perceived Susceptibility | Survey, SEM |
| 122 | 2022 | Lim, X. J., Ngew, P., Cheah, J. H., Cham, T. H., Liu, Y. | Go Digital: Can the Money-Gift Function Promote the Use of E-Wallet Apps | Survey, SEM |
| 123 | 2022 | Ojo, A.O., Fawehinmi, O., Ojo, O.T., Arasanmi, C., Tan, C.N.L. | Consumer Usage Intention of Electronic Wallets during the COVID-19 Pandemic in Malaysia | Survey, Partial Modelling Analysis |
| 124 | 202022 | Ming, K.L.Y., Jais, M. | Factors Affecting the Intention to Use E-Wallets During the COVID-19 Pandemic | Survey, SEM |
| 125 | 202022 | Al-Qudah, A. A., Al-Okaily, M., Alqudah, G., Ghazlat, A. | Mobile Payment Adoption in the Time of the COVID-19 Pandemic | Survey, SEM |
| 126 | 2022 | Shekhar, R., Jaidev, U.P. | Intention to Use Mobile Wallets: An Application of the Technology Acceptance Model | Survey, SEM |
| 127 | 2022 | Bailey, A. A., Bonifield, C. M., Arias, A., Villegas, J. | Mobile Payment Adoption in Latin America | Survey, SEM |
| 128 | 2022 | Bommer, W.H., Rana, S., Milevoj, E. | A Meta-Analysis of Ewallet Adoption Using the UTAUT Model | Meta Analysis, Relative Weight Analysis |

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
