# Peer review of "How Do Mobile Wallets Improve Sustainability in Payment Services? A Comprehensive Literature Review"

_sustainability, doi:10.3390/su142416541_

Round 1

Reviewer 1 Report

I would like to thank the editor to select me as a reviewer for this paper. After reviewing, I find that th paper attempts to examine “How Do Mobile Wallets Improve Sustainability in Payment Services? A Comprehensive Literature Review”. I find that this paper has a good analysis and it shows some major findings. In addition, the paper also has lots of discussions and show how mobile wallets improve sustainability, it is acceptable.

For better contribution to the literature, I have some revisions that are good for enhancing the quality of the manuscript.

1.      Section 1 should summarize some findings.

2.      The quality of Figure 1 is quite poor. Source for Figure 2

3.      The discussions should be further developed, especially the analysis of method, the most common findings and other weaknesses.

4.      The paper is lack of conclusions, and the limitations.

Thank you

Reviewer 2 Report

The article addresses an interesting topic and it is obvious that much time and effort was put into this work, and I appreciate it. However, I have some observations, as follows:

1. The abstract is wordy but says little about the methodology and contributions. Add some specifics.

2. In lines 220-227, there are too few articles in number of 128 to review the research problems of mobile wallets.

3. In lines 232-245, it is not clear that how to get the research topics into these 4 groups, the categorization need to be stated clearly.

4. In lines 232-245, as for the group of security & privacy, there are too few articles. I suggest that authors consider whether keep this group or not, In section 4, 4.4 should have been to focus on this group of security and privacy, but actually not.

5. In section 4.1 technology and adoption, it is much longer than 4.2 and 4.3, and the research structure and logic are confusing, I suggest to separate 4.1 into 4.1.1, 4.1.2 and 4.1.3 to show research structure clearly.

6. As a review paper, the future studies should be stated sufficiently. However, future studies only could be found in lines 732-726, those are not enough and need to be fully addressed.

7. Make a grammar check; there are some minor grammar problems.

Round 2

Reviewer 1 Report

I would like to thank the editor to select me as a reviewer for this paper. After reviewing, I think that the revised paper is good, and should be considered for publication.

Therefore, I made the acceptance on this paper.

Thank you

Reviewer 2 Report

It seems that the revised manuscript has been improved. So I suggest to accept it in present form.